# A General, Label-Free and Homogeneous Electrochemical Strategy for Probing of Protease Activity and Screening of Inhibitor

**DOI:** 10.3390/mi13050803

**Published:** 2022-05-21

**Authors:** Yunxiao Feng, Gang Liu, Fan Zhang, Jianwen Liu, Ming La, Ning Xia

**Affiliations:** 1College of Chemistry and Chemical Engineering, Pingdingshan University, Pingdingshan 467000, China; 2743@pdsu.edu.cn; 2College of Chemistry and Chemical Engineering, Henan University of Technology, Zhengzhou 450001, China; liugang08215@163.com; 3College of Chemistry and Chemical Engineering, Anyang Normal University, Anyang 455000, China; zzzfan@163.com (F.Z.); l718372708@163.com (J.L.)

**Keywords:** protease, electrochemical biosensor, copper, angiotensin-converting enzyme, thrombin

## Abstract

Proteases play a critical role in regulating various physiological processes from protein digestion to wound healing. Monitoring the activity of proteases and screening their inhibitors as potential drug molecules are of great importance for the early diagnosis and treatment of many diseases. In this work, we reported a general, label-free and homogeneous electrochemical method for monitoring protease activity based on the peptide–copper interaction. Cleavage of peptide substrate results in the generation of a copper-binding chelator peptide with a histidine residue in the first or third position (His^1^ or His^3^) at the N-terminal. The redox potential and current of copper coordinated with the product are different from the free copper or the copper complex with the substrate, thus allowing for the detection of protease activity. Angiotensin-converting enzyme (ACE) and thrombin were determined as the model analytes. The label-free and homogeneous electrochemical method can be used for screening protease inhibitors with high simplicity and sensitivity.

## 1. Introduction

Proteases play a vital role in numerous fundamental biological processes, including cell differentiation and growth, protein digestion, wound healing and activation of immune systems [1]. They can catalyze the hydrolysis of amide bonds in peptide chains with specific amino acid sequences. Abnormal activities of proteases have a tight relationship with many widespread diseases, such as Alzheimer’s disease, cancers, diabetics and so on [2,3]. Regulation of protease activities by potential drug molecules has proven to be a promising strategy for treating protease-related diseases. Therefore, extensive efforts have been devoted to developing simple, sensitive and low-cost methods for the determination of protease activity and screening of the potential inhibition drugs [2,3].

Protease-mediated cleavage can result in the fragmentation of substrate peptides into two shorter peptide chains or amino acids, which can be monitored by different techniques. Based on the detection format, protease activity assays can be divided into homogeneous analysis and heterogeneous analysis. Traditional methods, such as enzyme-linked immunosorbent assay (ELISA), gel electrophoresis and high-performance liquid chromatography (HPLC), always involve tedious separation/washing steps, complicated equipment and/or low sensitivity for protease detection [2,3]. To overcome these shortcomings, various novel methods have been developed recently, including mass spectrometry, fluorescence, colorimetric assays, quartz crystal microbalance, surface plasma resonance (SPR), surface-enhanced Raman scattering (SERS), and electrochemical, photoelectrochemical or electrochemiluminescent assays [2,3,4,5,6,7,8]. As a typical homogeneous assay, fluorescence methods have been broadly applied because of their distinguish merits of high sensitivity and fast response. However, the careful selection of the quencher/fluorophore pair and the expensive conjugating of the substrate onto both ends limit the applications, especially in resource-poor laboratories and regions. Electrochemical biosensors have attracted widespread attention because of their excellent advantages of high sensitivity, simplicity and instrument miniaturization. For protease detection, most of the electrochemical methods are heterogeneous and require the immobilization of peptide substrate on the electrode surface [9,10,11]. Hydrolysis of the peptide tethered on the electrode surface can lead to the departure of peptide fragments from the surface, thus affecting the electron transfer between the electrode and the electroactive species in solution. However, the small size of the hydrolysis product can only generate a slight fluctuation in terms of current intensity. For this view, extra electroactive molecules and nanomaterials are generally utilized to label the peptide substrate for signal output with well-defined electrochemical redox peaks [10]. Despite the enhanced sensitivity, the electrode modified with redox-labeled peptide may suffer from obvious signal loss after long-term preservation [12]. Moreover, in heterogeneous assays, the access of substrate peptide immobilized on the electrode surface to the catalytic center of protease may be relatively blocked due to the steric effect [4,7,13]. Thus, immobilization-free homogeneous electrochemical biosensors have aroused wide interests for protease detection. In the previous reports for homogeneous electrochemical assays of protease, the peptide substrate was usually labeled with a *p*-nitroaniline (pNA) group [14,15]. Cleavage of the peptide led to the release of electroactive pNA, thus allowing for the quantification of some proteases such as caspase-3 and trypsin. Nevertheless, the method involves complicated labeling procedures and is limited to the detection of other proteases. Thus, a general, label-free and homogeneous electrochemical strategy for monitoring of protease activity is still required.

Proteins and peptides can regulate the homeostasis of copper ions in organisms by precise coordination with different geometries and affinities in physiological conditions. The copper ion-anchoring proteins always contain histidine (His) residues located in different positions in the primary sequence. It has been demonstrated that the number and position of His residues have a significant influence on the coordination modes of copper ion-protein complexes [16,17,18,19]. Especially, the location of His residue in the first three N-terminal positions (His^1^, His^2^, and His^3^) can alter the metal binding affinity of the peptide chain and the catalytic activities of copper [17,20,21]. For this consideration, the amino-terminal Cu and Ni-binding (ATCUN) motif with the general sequence of H_2_N–Xxx-Zzz-His (XZH) at the N-terminus can be introduced into proteins and peptides for anticancer or antimicrobial applications [20,22,23,24]. For instance, beta-secretase can enzymatically transform a prochelator peptide into a high-affinity copper chelator peptide with the ATCUN motif at the N-terminus for stripping copper ions and suppressing the generation of copper-initiated reactive oxygen species [25,26]. In this work, we investigated the redox behavior of copper complexes performed with the peptide substrate and hydrolysis product with a His^1^ or His^3^ residue in the N-terminal. Based on the change in the redox potential and current of copper, the proteolytic hydrolysis of the peptide could be monitored, thus allowing for the development of a general, label-free and homogeneous electrochemical strategy for probing protease activity and screening of the inhibitor. To evaluate the analytical performance and demonstrate the application, angiotensin-converting enzyme (ACE) and thrombin were determined as the model analytes.

## 2. Materials and Methods

### 2.1. Materials

ACE and N-benzoyl-Gly-His-Leu (NBz-GHL) were obtained from Sigma-Aldrich (Shanghai, China). Thrombin was provided by Shanghai Yuanye Bio-Technology Co., Ltd. (Shanghai, China). The inhibitors and other chemicals were purchased from Shanghai Aladdin Biochemical Technology Co., Ltd. (Shanghai, China). The peptides were ordered from ChinaPeptides Co., Ltd. (Shanghai, China). All the solutions were prepared freshly using ultrapure deionized water (>18 MΩ·cm).

### 2.2. Instruments

The electrochemical measurements were conducted on a CHI 660E electrochemical workstation (Shanghai, China) in a homemade plastic cell. A cleaned glassy carbon electrode, a platinum wire and a Ag/AgCl were used as the working electrode, auxiliary electrode and reference electrode, respectively.

### 2.3. Analysis of ACE

ACE was dissolved in deionized water and then diluted to the desired concentration with Tris buffer (100 mM, pH 8.2) containing 200 mM NaCl. For the assay of ACE, 10 μL of NBz-GHL substrate at a fixed concentration was mixed with 10 μL of ACE at a known concentration. The mixture was then incubated at 37 °C for 60 min to ensure the hydrolysis of the NBz-GHL substrate. To evaluate the inhibition efficiency of the inhibitor, ACE was pre-incubated with a given concentration of enalaprilat at room temperature for 10 min and then incubated with the substrate. After the proteolytic reaction, 20 μL of Cu(II)-containing Tris buffer was added to the mixture and then analyzed by differential pulse voltammetry. The amplitude is 0.05 V and the pulse width is 0.05 s.

### 2.4. Analysis of Thrombin

The thrombin powder was dissolved in the Tris buffer at a high concentration. Then, 10 μL of a peptide substrate (GARGGH) at a fixed concentration was mixed with 10 μL of diluted thrombin at different concentrations. The mixture was then incubated at 37 °C for 1 h to ensure the hydrolysis of GARGGH. To evaluate the inhibition efficiency of the inhibitor, thrombin was pre-incubated with a given concentration of argatroban at room temperature for 10 min and then incubated with the substrate. After the proteolytic reaction, 20 μL of Cu(II)-containing Tris buffer was added to the mixture and then analyzed by linear sweep voltammetry with a scan rate of 100 mV/s.

## 3. Results and Discussion

### 3.1. Principle of the Proposal

Histidine residue in the peptide chain plays a decisive role in the binding of copper ions. For example, the peptide with a His^1^ motif in the N-terminal can coordinate with Cu(II) in a non-saturating binding format [16,17]. Our group found that the peptide with a His^3^ motif in the N-terminal position (known as ATCUN) shows a unique Cu(II)-binding property and redox activity. The ATCUN-Cu(II) can be oxidized into ATCUN-Cu(III) at around 0.8 V and thus exhibits an ability for electrocatalytic water oxidation [21,27,28]. The proteolytic removal of a propeptide residue can lead to the presence of a His^1^, His^2^ or His^3^ motif in the N-terminal. ACE, a class of zinc-dependent peptidases, plays a major physiological role in regulating blood pressure in a renin–angiotensin system. NBz-GHL, the commercial substrate of ACE, can be proteolytically digested to produce a His^1^-containing peptide HL. Herein, ACE was first determined as the model analyte (Figure 1A). To demonstrate the feasibility and versatility of the label-free electrochemical strategy, thrombin (a serine protease) was also determined with a peptide of GARGGH as the substrate (Figure 1B). The peptide can be proteolytically cleaved by thrombin between the Arginine (R) and Glycine (G) residues to produce a Cu(II)-binding peptide GGH. The resulting copper complexes (Cu(II)-HL and Cu(II)-GGH) exhibit different redox properties with the free Cu(II) or Cu(II)-substrate complex, thus allowing for the detection of protease activity.

### 3.2. Electrochemical Behavior of Copper Complexes

To probe the feasibility of our proposal, we first investigate the electrochemical behavior of Cu(II) in the presence of different peptides. As shown in Figure 2A, the cyclic voltammogram of Cu(II)/NBz-GHL exhibited a couple of quasi-reversible redox waves in the air or nitrogen-saturated Tris buffer, which is similar to that of free Cu(II) (data not shown). For Cu(II)/HL, an irreversible CV wave was observed in the air-saturated solution. The CV became quasi-reversible when the solution was saturated with nitrogen. Thus, the irreversible sigmoidal redox wave in the air-saturated solution should be attributed to the catalytic reaction in which Cu(II) is immediately regenerated by O_2_ during the electrochemical reduction process [29]. The redox potential and activity of Cu(II)/HL are obviously different from those of Cu(II)/NBz-GHL. The result is acceptable since Cu(II) can be coordinated by the imidazole group in histidine residue and the free amine in the N-terminal of the dipeptide. As expected, the CV of Cu(II)/GGH exhibited an irreversible redox wave which is apparently distinguished from that of Cu(II)/GARGGH (Figure 2B). This is attributed to the electrocatalytic water oxidation by the Cu(II)/GGH complex [21,27,28]. These results demonstrate that the electrochemical behavior of the Cu(II) complexes formed with the proteolytic products is essentially distinguished from that of free Cu(II) in the presence of substrate. Thus, the activity of protease (ACE and thrombin) may be monitored by the histidine-regulated binding of Cu(II).

### 3.3. Feasibility of the Method

Differential pulse voltammetry and linear sweep voltammetry are more sensitive than cyclic voltammetry due to their low background current. To investigate the analytical performance of the method, the two electrochemical techniques were used to monitor the signal change. As shown in Figure 3A, the reduction potential (E_pc_) of Cu(II)/NBz-GHL is 0.038 V (curve 1), and it shifts to around −0.017 V in the Cu(II)/HL solution (curve 2). When NBz-GHL was incubated with a given concentration of ACE, the reduction current (*I*_pc_) at 0.038 V decreased (curve 3). This is accompanied by the presence of a new peak at −0.017 V, which is attributed to the formation of Cu(II)/HL. Interestingly, when NBz-GHL was incubated with the mixture of ACE and its inhibitor (enalaprilat) (curve 4), the current at 0.038 V was higher but the *I*_pc_ at −0.017 V was significantly lower than that in the absence of the inhibitor. This indicated that inhibition of the ACE activity depressed the generation of HL, thus leading to the decrease in the *I*_pc_ of Cu(II)/HL. The result demonstrated that the label-free peptide substrate can be used as the probe to monitor the activity of ACE.

Figure 3B depicts the LSV curves for Cu(II) in the presence of GARGGH, GGH. GARGGH/ACE, and the mixture of GARGGH, thrombin as well as inhibitor. No oxidation wave was observed for Cu(II)/GARGGH (curve 1), which is indicative of the low background signal of the method. When GARGGH was incubated with a given concentration of thrombin, a catalytic oxidation wave was observed (curve 3), which is attributed to the oxidation of Cu(II)/GGH (curve 2). The oxidation peak current *I*_pa_ was decreased when GARGGH was incubated with the mixture of thrombin and its inhibitor (argatroban) (curve 4), indicating that the generation of Cu(II)/GGH could be prevented by inhibiting the activity of thrombin. Thus, the label-free method can be applied to probing thrombin activity and screening its potential inhibitors.

### 3.4. Sensitivity

To evaluate the sensitivity, the peptide substrate was incubated with different concentrations of protease (ACE or thrombin), followed by the addition of Cu(II). Figure 4A depicts the results for the determination of different concentrations of ACE. It can be seen that the *I*_pc_ at 0.038 V decreased with the increase of ACE concentration in the range of 0~200 mU/mL, which is accompanied by the increase in the reduction current at −0.017 V. The linear relationship between ACE concentration and *I*_pc_ at 0.038 V was *I*_pc_ = 0.35 − 0.004 [ACE] (mU/mL) (Figure 4B). The detection limit was estimated to be 1 mU/mL, which is comparable to that achieved by HPLC, mass spectrometry, fluorescence and electrophoresis [30,31,32,33]. However, the proposed electrochemical method did not require the use of complicated and expensive equipment.

Figure 5A shows the results for the detection of different concentrations of thrombin. It can be observed that the *I*_pa_ was intensified with the increase in thrombin concentration. A linear curve was obtained in the concentration range of 0.5~250 mU/mL (Figure 5B). The *I*_pa_ began to level off beyond 250 mU/mL. The platform is indicative of the digestion of most peptide substrates. We found that increasing the concentration of peptide substrate and Cu(II) to 200 μM did not significantly affect the sensitivity, but could expand the linear dynamic range of the system. The linear equation can be expressed as *I*_pa_ = 0.01 [thrombin] (mU/mL) + 0.134 with a detection limit down to 0.5 mU/mL. The high sensitivity can be attributed to the excellent electrocatalytic property of the Cu(II)-GGH complex for water oxidation and the low background current of the method.

### 3.5. Inhibition Assays

The screening of protease inhibitors is beneficial for the discovery of potential drugs to treat many diseases. Thus, the electrochemical method was employed to assay the inhibition efficiency of two well-known inhibitors enalaprilat and argatroban for ACE and thrombin, respectively. Consequently, when ACE at a concentration of 50 mU/mL was incubated with different concentrations of enalaprilat, the current at 0.038 V was enhanced with the increase of inhibitor concentration. This suggested that the activity of ACE can be well inhibited by enalaprilat. Figure 6A shows the dependence of inhibition efficiency on enalaprilat concentration. The half-maximal inhibitory concentration (IC_50_) was found to be 78 nM, which is consistent with that obtained by other methods [34]. We also monitored the current change for measuring a fixed concentration of thrombin in the presence of different concentrations of argatroban. It was found that the current decreased with the increase of argatroban concentration. The relationship between inhibition efficiency and enalaprilat concentration was shown in Figure 6B. The IC_50_ of argatroban for 100 mU/mL thrombin was calculated to be 0.29 μM based on the change of inhibition efficiency. The result is in agreement with that achieved by other methods [35,36].

## 4. Conclusions

In summary, we proposed a general and label-free electrochemical method for the detection of protease. It was based on the difference in the redox potential and current of copper coordinated by the substrate and proteolytic product. The analytical performances were demonstrated with ACE and thrombin as the model analytes. The detection limits for ACE and thrombin detection were 1 and 0.5 mU/mL, respectively. The method was used to evaluate the inhibition efficiency of well-known inhibitors with satisfactory results. We believe that the work is valuable for the design of novel biosensors and the discovery of effective inhibitor drugs.

## Figures and Tables

**Figure 1 micromachines-13-00803-f001:**
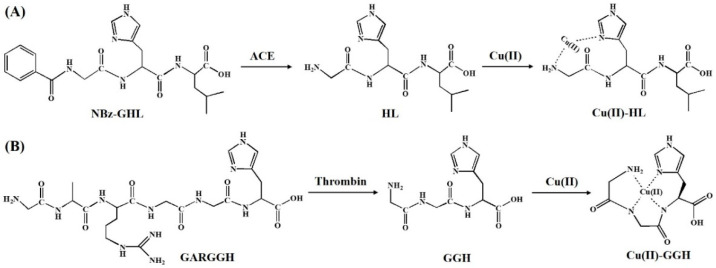
Schematic representation for proteolytic reactions and Cu(II) complexes.

**Figure 2 micromachines-13-00803-f002:**
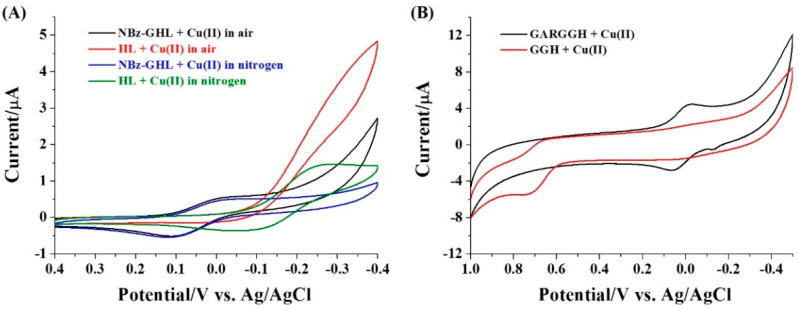
CVs of Cu(II) in air or nitrogen-saturated solution with the addition of different peptides: (**A**) NBz-GHL and HL; (**B**) GARGGH and GGH. The final concentrations of peptide and Cu(II) were 100 μM. The scan rate in panel A is 20 mV/s and that in panel B is 100 mV/s.

**Figure 3 micromachines-13-00803-f003:**
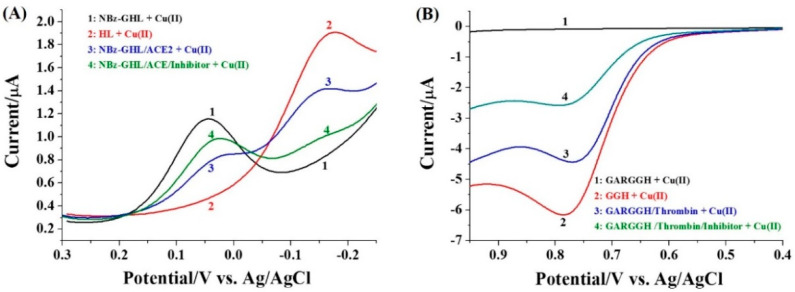
Differential pulse voltammograms (DPVs) (**A**) and linear sweep voltammograms (LSVs) (**B**) of Cu(II) in the presence of different substances. The concentrations of peptide and Cu(II) were 100 μM. The proteolytic reaction time was 30 min. The proteolytic reaction time was 30 min. In panel A, the concentrations of ACE and inhibitor enalaprilat were 250 mU/mL and 0.5 μM, respectively. In panel B, the concentrations of thrombin and inhibitor argatroban were 500 mU/mL and 1 μM, respectively.

**Figure 4 micromachines-13-00803-f004:**
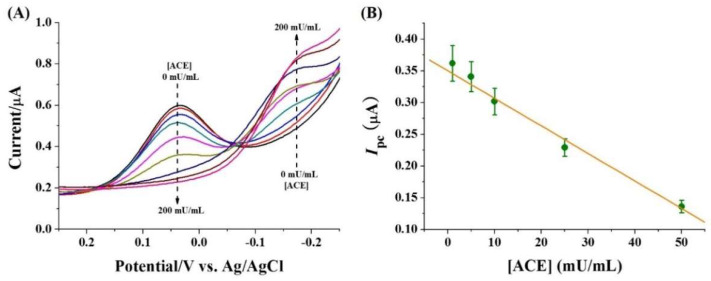
(**A**) DPVs for the detection of different concentrations of ACE (0, 1, 5, 10, 25, 50, 100 and 200 mU/mL) and (**B**) calibration plots for the detection of ACE in the range of 1~50 mU/mL. The concentrations of substrate and Cu(II) were 50 μM.

**Figure 5 micromachines-13-00803-f005:**
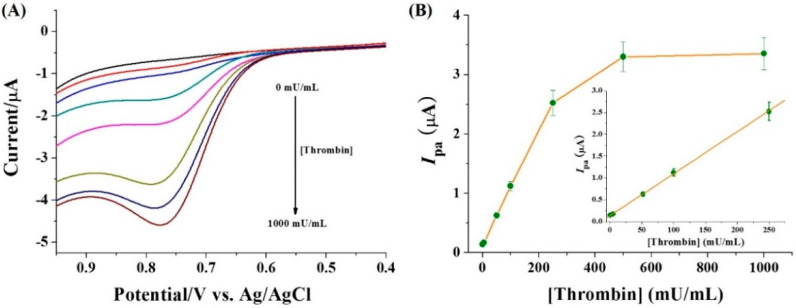
LSVs (**A**) and calibration plots (**B**) for the detection of different concentrations of thrombin (0, 0.5, 5, 50, 100, 250, 500 and 1000 mU/mL). The inset shows the linear portion of the calibration plots. The concentrations of peptide substrate and Cu(II) were 100 μM.

**Figure 6 micromachines-13-00803-f006:**
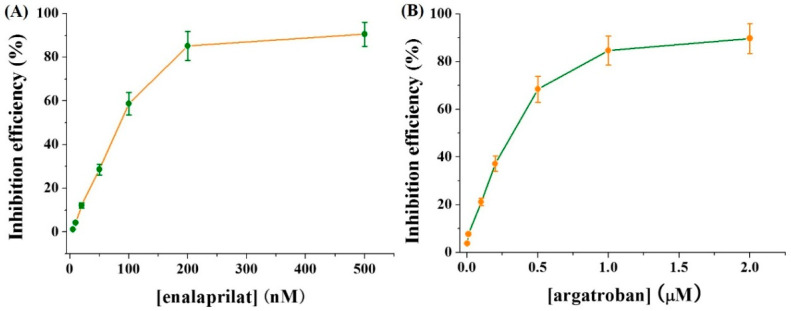
Effect of enlaprilat (**A**) and argatroban (**B**) concentration on the inhibition efficiency. The concentrations of ACE and thrombin were 50 and 100 mU/mL, respectively.

## Data Availability

Not applicable.

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
