# Peer review of "A General, Label-Free and Homogeneous Electrochemical Strategy for Probing of Protease Activity and Screening of Inhibitor"

_micromachines, 2022, doi:10.3390/mi13050803_

Round 1
Reviewer 1 Report
In this manuscript, authors described a kind of label-free electrochemical method to probe the protease activity and screening of inhibitor, which is simple and interesting. However, several points are missing and not so clear:
1.It's not clear if NBz-GHL will react with Cu(II). The cylic voltametry of pure Cu(II) without any analyte is not given. And there are no other characterization methods to further support the proposed coordination (or not) reaction between Cu(II) and the analyte.
2.On Page4, line155, the author explained that the irreversible peak was due to Cu(II) regeneration, but usually electrochemical O2 reduction occurred around this potential, H2O2 will be formed. So the explaination in this part need to be reconsidered.
3. I suppose most of the CVs in this manuscript are carried out under nitrogen, but it was not mentioned either in experimental part or figure caption. The scan rate for all the electrochemical techniques are not given.
4. On page6, line214, "Figure 4B" should be changed to "Figure 5".
5. The concentration of substrate and Cu(II) might influce the sensitivity of this system, but it was not discussed in the manuscript.
Reviewer 2 Report
The authors developed an electrochemical method to detect protease based on the Cu(II)-histidine binding affecting the redox responses. It is a novel, label-free method to detect protease. The article is well-organized and well written, yet I found some experimental details are missing and some electrochemical understanding of the system could be improved. Here are my comments:
- The authors could add more information about their electrochemical experiment, for example, what is the scan rate in the CV and LSV, etc?
- In Fig. 5B, the response is linear with respect to [Thrombin] up to 250 mU/mL. The authors did not stress this and simply gave the equation of the linear fit. This is not accurate and misleading. Also, the authors should explain why the response are not linear at higher concentrations of Thrombin.
- Is there any reason why DPV is used for the detection of ACE and LSV is used for Thrombin? Why not use DPV for both species, or LSV for both?
- It will be better if the authors could explain more about the CVs in Fig. 2, and explain it more quantitatively. For example, in Fig. 2B, what is the pair of peaks around 0 V in the CV of GARGGH+Cu(II)? It seems to be the redox of Cu(II). Then why is the peak current different from that in FIg. 2A (NBz-GHL +Cu(II))? The concentrations of Cu(II) in both cases are the same.
- The authors discussed about the irreversible peak in HL+Cu(II) in air solution. I kind of agree with their explanation, but can they provide more information on the reaction. For example, what is Cu(II)-HL reduced to, and how does O2 oxidize it. Or maybe the authors can add citations about the mechanism.
- A minor issue on the use of 'reversible' in line 154. When describing CVs, a reversible CV usually means a CV with Nerstian behavior, i.e. narrow peak separation and fast kinetics. A CV with slow kinetics can still show a peak in the forward scan and a peak in the reverse scan, but it is not called 'reversible'. I know the authors meant that a peak appeared on the reverse scan, but they should be careful with the expressions.
Author Response
We thank the reviewer for his/her positive comments: “The authors developed an electrochemical method to detect protease based on the Cu(II)-histidine binding affecting the redox responses. It is a novel, label-free method to detect protease. The article is well-organized and well written, yet I found some experimental details are missing and some electrochemical understanding of the system could be improved. Here are my comments:”
Comment 1: “The authors could add more information about their electrochemical experiment, for example, what is the scan rate in the CV and LSV, etc?”
Response: We have added the details for the electrochemical experimental, including the experimental conditions and electrochemical parameters.
Comment 2: “In Fig. 5B, the response is linear with respect to [Thrombin] up to 250 mU/mL. The authors did not stress this and simply gave the equation of the linear fit. This is not accurate and misleading. Also, the authors should explain why the response are not linear at higher concentrations of Thrombin.”
Response: We have added the comments to discuss the results on Page 6. The responses are not linear at higher concentrations of thrombin because most of peptide substrates have been digested. Thus, the dynamic linear range could be expanded by increasing the concentration of peptide substrate and Cu(II).
Comment 3: “Is there any reason why DPV is used for the detection of ACE and LSV is used for Thrombin? Why not use DPV for both species, or LSV for both?”
Response: To demonstrate the feasibility and universality of our proposal, two electrochemical techniques were used. The users can choose one of them to determine proteases by a standard curve method.
Comment 4: “It will be better if the authors could explain more about the CVs in Fig. 2, and explain it more quantitatively. For example, in Fig. 2B, what is the pair of peaks around 0 V in the CV of GARGGH+Cu(II)? It seems to be the redox of Cu(II). Then why is the peak current different from that in FIg. 2A (NBz-GHL +Cu(II))? The concentrations of Cu(II) in both cases are the same.”
Response: The peptide substrate of GARGGH shows poor binding with Cu(II). There is no essential difference in the redox potential of Cu(II) before and after the addition of peptide substrate. We have written the sentence in the manuscript: “the cyclic voltammogram of Cu(II)/NBz-GHL exhibited a couple of reversible redox waves in the air or nitrogen-saturated Tris buffer, which is similar to that of free Cu(II) (data not shown).” The difference in the peak current between GARGGH/Cu(II) and NBz-GHL/Cu(II) can be attributed to the difference in the scan rate. We have added the details for the experimental conditions and electrochemical parameters.
Comment 5: “The authors discussed about the irreversible peak in HL+Cu(II) in air solution. I kind of agree with their explanation, but can they provide more information on the reaction. For example, what is Cu(II)-HL reduced to, and how does O2 oxidize it. Or maybe the authors can add citations about the mechanism.”
Response: We have discussed the mechanism on Page 4 and cited the reference (J. Am. Chem. Soc. 2011, 133, 12229–12237) to support the comment.
Comment 6: “A minor issue on the use of 'reversible' in line 154. When describing CVs, a reversible CV usually means a CV with Nerstian behavior, i.e. narrow peak separation and fast kinetics. A CV with slow kinetics can still show a peak in the forward scan and a peak in the reverse scan, but it is not called 'reversible'. I know the authors meant that a peak appeared on the reverse scan, but they should be careful with the expressions.”
Response: It is a good suggestion. We have changed “reversible” into “quasi-reversible”.
Reviewer 3 Report
The manuscript describes work which is both novel and interesting as well as being technically sound and well described. I am happy to recommend publication following attention to the relatively minor points below.
In Fig. 1 it would be very interesting to know if there is evidence for the proposed copper complexes with the product peptides.
In Fig. 3 and associated text it would be useful to know the time-scale of the electrochemical measurements and whether real-time work or enzyme kinetic studies might be possible. Some more details of the instrumentation itself would be very useful.
In Fig. 6 and text it would be very useful and important to know how the measured IC50 values compare with those determined by other methods.
Author Response
We thank the reviewer for his/her positive comments: “The manuscript describes work which is both novel and interesting as well as being technically sound and well described. I am happy to recommend publication following attention to the relatively minor points below.”
Comment 1: “In Fig. 1 it would be very interesting to know if there is evidence for the proposed copper complexes with the product peptides.”
Response: The formation of copper complexes can be evidenced by the change in the redox potential of Cu(II). This method has been widely used to demonstrate the formation of peptide-copper complexes (J. Am. Chem. Soc. 2011, 133, 12229–12237). Moreover, the peptide with a His residue in the first position of N-terminal can coordinate with Cu(II) in a non-saturating binding format. We have cited the references (Chem. Biodiversity 2021, 18, e2100043; Coord. Chem. Rev. 2016, 327–328, 43) to support the statement.
Comment 2: “In Fig. 3 and associated text it would be useful to know the time-scale of the electrochemical measurements and whether real-time work or enzyme kinetic studies might be possible. Some more details of the instrumentation itself would be very useful.”
Response: It is an excellent comment. Considering that Cu2+ may affect the activity of protease, Cu2+ was added into the peptide/protease solution for signal readout after the completion of enzymatic hydrolysis reaction.
Comment 3: “In Fig. 6 and text it would be very useful and important to know how the measured IC50 values compare with those determined by other methods.”
Response: It is a good question. The value is consistent with that obtained by other methods. We have added the comments and cited the references in the revised manuscript.